# Scaling (Down) CLIP: A Comprehensive Analysis of Data, Architecture, and Training Strategies

**Zichao Li** *                                                                      *zli489@ucsc.edu*
*Google Deepmind*
*University of California, Santa Cruz*

**Cihang Xie**                                                                       *cixie@ucsc.edu*
*University of California, Santa Cruz*

**Ekin Dogus Cubuk**                                                                 *cubuk@google.com*
*Google Deepmind*

**Reviewed on OpenReview:** *https://openreview.net/forum?id=t4nnCi5AO6*

## Abstract

This paper investigates the performance of the Contrastive Language-Image Pre-training (CLIP) when scaled down to limited computation budgets. We explore CLIP along three dimensions: data, architecture, and training strategies. With regards to data, we demonstrate the significance of high-quality training data and show that a smaller dataset of high-quality data can outperform a larger dataset with lower quality. We also examine how model performance varies with different dataset sizes, suggesting that smaller ViT models are better suited for smaller datasets, while larger models perform better on larger datasets with fixed compute. Additionally, we provide guidance on when to choose a CNN-based architecture or a ViT-based architecture for CLIP training. We compare four CLIP training strategies - SLIP, FLIP, CLIP, and CLIP+Data Augmentation - and show that the choice of training strategy depends on the available compute resource. Our analysis reveals that CLIP+Data Augmentation can achieve comparable performance to CLIP using only half of the training data. This work provides practical insights into how to effectively train and deploy CLIP models, making them more accessible and affordable for practical use in various applications.

## 1 Introduction

In recent years, there has been a surge of interest in image-and-language representation learning (Radford et al., 2021; Jia et al., 2021; Pham et al., 2021; Ramesh et al., 2022), which aims to capture the rich and complex interplay between visual and textual information. One of the most promising approaches in this area is the Contrastive Language-Image Pre-Training (CLIP) framework, which leverages large-scale text and image data to learn a unified representation space for both modalities. CLIP has achieved state-of-the-art performance on a range of downstream tasks and has been shown to generalize well to out-of-distribution data (Chen et al., 2019; Jia et al., 2021; Miller et al., 2021; Taori et al., 2020; Nguyen et al., 2022; Gontijo-Lopes et al., 2021).

While previous studies on scaling Contrastive Language-Image Pre-Training (CLIP), such as Li et al. (2022); Cherti et al. (2022), have primarily focused on large-computation scenarios, our work aims to explore the performance of CLIP under resource constraints. We present a comprehensive study of scaling down CLIP in three directions: data, architecture, and training strategies. Specifically, we investigate the effect of using different training data sizes, compare the performance of various architectures at different computation

---

*Work done during an internship at Google.

budgets, and evaluate the effectiveness of different pre-training strategies. Our experiments are conducted on a large image-and-language dataset, WebLI (Chen et al., 2022), which contains over 3.4 billion image-text pairs in English. Particularly, we set a computation limit for most of our experiments, and the total number of sampled data for most experiments is **at most 3.4B**.

We train on datasets of different sizes to explore if the performance on ImageNet variants align with the performance on ImageNet. We also explore the importance of data quality and demonstrate that a smaller set of high-quality data can outperform larger datasets. We find that using only the higher quality 40% of the data can achieve better performance than using the entire dataset, and we investigate how model performance changes with increasing dataset sizes. Our results provide guidance on how to select training data for CLIP models, a critical issue for practical applications.

Concerning architecture, we compare multiple architectures under various dataset sizes. We investigate how to choose size of the vision encoder depending on the dataset size and compute budget. We show that the a larger ViT (Dosovitskiy et al., 2020) is not always better. We also demonstrate the importance of selecting between CNN (He et al., 2016) and ViT architectures for CLIP training. Although previous studies have shown that ViT-based CLIP has better performance than CNN-based CLIP models, we find that when the number of sampled data is small, CNNs perform better.

Finally, we compare four options: SLIP (Mu et al., 2021), FLIP (Li et al., 2022), CLIP, and CLIP+Data Augmentation. We show that SLIP Mu et al. (2021) is not always the best choice compared to CLIP and FLIP. When the size of the training set is small, SLIP performs better than CLIP. However, as the training data size increases, SLIP has similar performance to CLIP but incurs twice the computational cost. Our results provide insights into the trade-offs between computational cost and performance in CLIP training, a critical issue for practical applications.

Finally, in terms of training strategies, we compare four options: SLIP Mu et al. (2021), FLIP Li et al. (2022), CLIP, and CLIP+Data Augmentation. We show that SLIP Mu et al. (2021) does not always outperform CLIP and FLIP. When the size of the training set is small, SLIP performs better than CLIP. However, at larger training set sizes, SLIP has a similar performance to CLIP, but requires twice the computational cost. We explore the trade-offs between computational cost and performance.

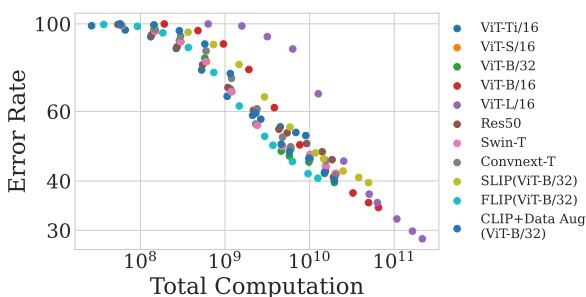
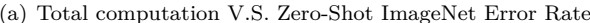
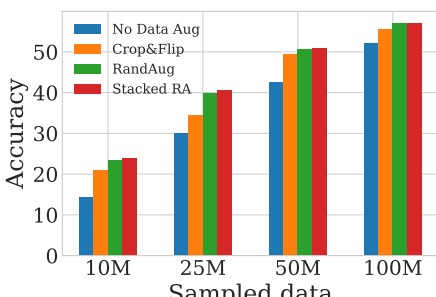

(a) Total computation V.S. Zero-Shot ImageNet Error Rate

(b) Comparison between various data augmentation for CLIP

Figure 1: **Training a High-Quality CLIP Model:** This figure highlights the main contributions of our work. In Figure 1(a), we demonstrate the relationship between different models, strategies, and error rates on the ImageNet dataset. The total computation is computed by GFLOPs per sample times the number sampled data. Additionally, in Figure 1(b), we illustrate how data augmentation methods improve the zero-shot performance of various datasets.

## 2 Related Work

**Contrastive Language-Image Pre-Training:** In recent years, the field of natural language processing (NLP) has made significant strides in pre-training language models on large amounts of text data (Peters

et al., 2018; Devlin et al., 2019; Radford et al., 2018). Concurrently, computer vision has witnessed a surge in pre-training convolutional neural networks (CNNs) on large-scale image datasets (He et al., 2016; Huang et al., 2017; Simonyan & Zisserman, 2014). The CLIP model (Radford et al., 2021) is introduced as a joint pre-training framework for image and text representations. It utilizes a contrastive loss function to learn a shared embedding space between images and their corresponding textual descriptions. CLIP has demonstrated remarkable zero-shot performance on various visual classification tasks, achieving accuracy levels comparable to those obtained by supervised learning methods.

Recently, LiT (Zhai et al., 2021) has been proposed as an extension to CLIP. LiT introduces locked-image tuning, which adopts a pre-trained visual encoder and updates only the text model. As a result, LiT can attain better zero-shot performances more efficiently. The success of CLIP has also inspired the work of SLIP (Mu et al., 2021), which combines self-supervised learning with CLIP to enhance the visual encoder.

**Scaling Language-Image Pre-training:** In recent years, considerable effort has been devoted to enhancing the efficiency and scalability of the Contrastive Language-Image Pre-training (CLIP) method. To improve scalability, Li et al. (2022) has proposed a method called FLIP, which masks half or more patches of the training images to reduce computation by 2x and allow for the use of larger batch sizes, while maintaining the performance of the original CLIP models. In a similar vein, Cherti et al. (2022) has provided further insights into the scaling laws of CLIP models, revealing that the training distribution plays a pivotal role in scaling laws, as evidenced by the different scaling behaviors exhibited by OpenCLIP Ilharco et al. (2021) models, despite having identical model architectures and similar training procedures.

Research on scaling CLIP models has predominantly focused on training with large computational resources, often starting from a ViT large model and training on massive datasets over multiple epochs. For example, the largest number of sampled data in Cherti et al. (2022) is 34 billion. Less attention has been paid to the performance of CLIP models with smaller training budgets.

## 3 Methods

**Training Pipeline** We adopt the same training pipeline as CLIP (Radford et al., 2021), which employs a contrastive loss to train both the vision encoder and text encoder from scratch. The contrastive loss encourages the vision and text encoders to map corresponding image-text pairs to similar feature representations in a shared embedding space. Specifically, the contrastive loss minimizes the distance between the representations of positive pairs (i.e., image-text pairs that belong together) while maximizing the distance between the representations of negative pairs (i.e., image-text pairs that do not belong together).

**Dataset and Data Pre-processing** We conduct our experiments on WebLI Chen et al. (2022), a large image-and-language dataset built from the public web, containing over 10 billion image-text pairs from more than 100 languages. We specifically focus on pairs in the English language, resulting in a dataset of approximately 3.4 billion pairs. In our base experiments (referred to as just CLIP), we do not apply any data augmentation to images, except for resizing it to 224x224 and normalizing pixel values to the range of -1 to 1.

For text processing, we use a SentencePiece tokenizer with a vocabulary size of 32k. We set the token length to 16 and pad or truncate sentences to that length.

### 3.1 Hyper-parameters

In our study, we adopt the hyper-parameter settings used in a previous work Zhai et al. (2021). We use AdafactorShazeer & Stern (2018) as the optimizer with $\beta_1$ and $\beta_2$ set to 0.9 and 0.999, respectively, and set the batch size of all our models to 16k. To adjust the learning rate, we use a cosine learning scheduler with an initial rate of 0.001, and we set the weight decay to 0.0001. These hyper-parameters are carefully selected to ensure optimal performance of the CLIP model in our experiments.

### 3.2 Evaluation Metrics:

In the zero-shot transfer evaluation, we aim to assess the model's ability to generalize to new tasks without any fine-tuning. To ensure a fair comparison with prior work, we follow the same evaluation settings as Radford et al. (2021); Zhai et al. (2021). We use the same prompts collected from these works and pre-process the test images by first resizing them and then applying a central crop with a 0.875 aspect ratio to match the target resolution.

For the linear probe evaluations, we freeze the vision encoder and zero-initialize the fully connected layer. We then sweep the learning rate to find the best rate for each architectures.

For the evaluation of retrieval performance on MSCOCO captions, we report the results based on the test set. For image-to-text retrieval metrics, we rank all text captions in descending order based on the cosine similarity of their embeddings with the image embedding. Subsequently, we report the proportion of images that achieved the Recall@1 performance. For text-to-image retrieval, we follow a similar approach but rank the images instead and calculate the average results across all texts.

## 4 Data

**Data Quantity** To assess the influence of data quantity on performance, we conduct experiments using five datasets of varying sizes: 10M, 25M, 100M, 200M, and 400M. We choose the 400M as the largest dataset in this section as it is commonly used in CLIP training. For all CLIP models in this section, we use ViT-B/32 as the vision encoder. We train models for number of epochs ranging from 2 to 32.

The findings regarding the zero-shot performance of CLIP models on ImageNet are depicted in Figure 2(a). The experiments reveal that the impact of training data quantity on model performance is contingent upon both the dataset size and the number of training epochs. In particular, we note that for smaller datasets such as the 25M dataset, increasing the training epochs does not yield significant performance improvements on ImageNet. However, for larger datasets like the 400M dataset, more training epochs do result in enhanced performance on ImageNet.

Additionally, we evaluate the average zero-shot performance of our models on ImageNet variants, as illustrated in Figure 2(b). The average zero-shot performance across six ImageNet variants follows a similar trend to that observed on ImageNet: larger datasets and longer training result in improved performances. However, we observe that the performance trends on ImageNet variants do not consistently correspond to those on ImageNet for each specific variant, as shown in Figure 20 in Appendix. For instance, while increasing the number of epochs from 16 to 32 does not improve performance on ImageNet for the 25M dataset, it leads to improved performance on ImageNet-A. Similarly, our models trained for 32 epochs exhibit significantly better performance on ImageNet-R and ImageNet-Sketch compared to those trained for 16 epochs, despite no improvement on ImageNet.

Additionally, we conduct experiments on few-shot and retrieval scenarios, and the results pertaining to few-shot and retrieval performances across different dataset sizes are depicted in Figures 3 and 4. We observe that the few-shot performances demonstrate a similar trend to the zero-shot performances on ImageNet. However, we notice a slightly distinct trend in retrieval performances. Specifically, we note that when the number of epochs surpasses eight, there is minimal improvement in both image retrieval and text retrieval performances.

**Data Quality** We compute the similarity between images and texts to create subsets from the 3.4B dataset, containing the top 20%, 40%, 60%, and 80% of the highest quality data. These filtered datasets are then used to train models for a single epoch and evaluate their zero-shot performance on ImageNet. The findings from our study, as depicted in Figure 5(a), suggest that the number of samples encountered by the CLIP model does not always have the greatest influence on performance. Interestingly, despite significantly fewer iterations than the full dataset, the model trained on the Top40% subset demonstrates notably superior performance.

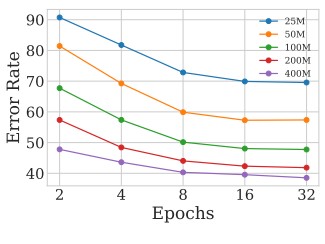

(a) Training Epochs V.S. ImageNet Accuracy

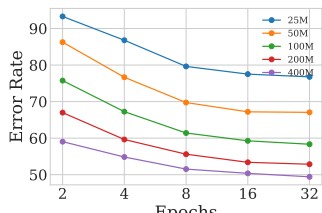

(b) Training Epochs V.S. Average ImageNet Variants Error Rate

Figure 2: **Data Quantity:** Zero-Shot performances with the same dataset size across varied training epochs

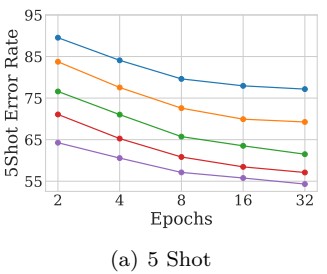

(a) 5 Shot

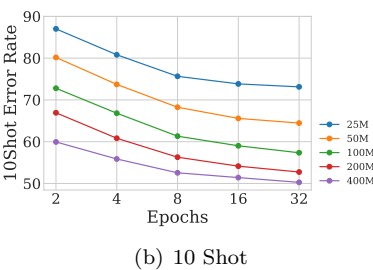

(b) 10 Shot

Figure 3: **Data Quantity:** Few-Shot Performances on ImageNet.

To ensure a fair comparison among datasets of varying quality, we also train datasets with an equal number of sampled data points, and the results are presented in Figure 5(b). Our findings suggest that, on ImageNet, the Top40% dataset achieves the highest performance, whereas the Top20% dataset falls short of the Top40% in terms of performance. These results emphasize the importance of data quality in training effective CLIP models and highlight the potential benefits of focusing on high-quality data subsets in this context.

We display the few-shot and retrieval performances on datasets with different qualities in Figure 6 and Figure 7, respectively. The results indicate that datasets with higher quality yield superior few-shot and retrieval performances, even with a relatively small number of sampled data points. Moreover, when the number of sampled data points is consistent, datasets of higher quality demonstrate superior 5-shot and 10-shot performances. Particularly, the top 80% dataset exhibits the most impressive retrieval performances.

## 5 Variants of Vision Transformers

In this section, our aim is to investigate how the performance of different CLIP models, which vary in the size of their vision encoders, is affected by dataset size and the number of sampled data. To achieve this goal, we present the zero-shot performance and linear probing results of these models, providing a more comprehensive understanding of the impact of dataset size and the number of data samples on CLIP's performance.

To investigate the effect of vision encoders on CLIP's performance, we fixed the size of the text transformers at vit-base and experimented with various vision encoders, including ViT-Ti/16, S/16, B/32, B/16, and L/16 (Dosovitskiy et al., 2020).

The literature on scaling up CLIP typically demonstrates a power law starting with a large computation cost. To evaluate CLIP's performance as we scale up the dataset size, we sampled ten subsets from the entire dataset, with sizes ranging from 10M to 3.4B. These subsets have the same data distribution and quality and are used in this section and subsequent sections. To assess the effect of the number of sampled data, we provide results for all subsets trained for one epoch. To ensure a fair comparison, we also provide results for all subsets trained for the same number of iterations, resulting in the same number of sampled data.

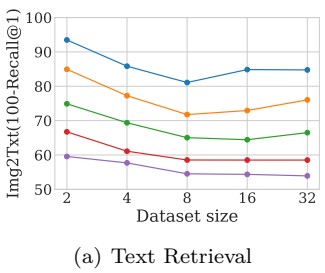

(a) Text Retrieval

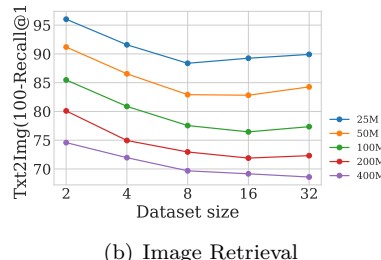

(b) Image Retrieval

Figure 4: **Data Quantity:** Retrieval Performances on MSCOCO (Chen et al., 2015).

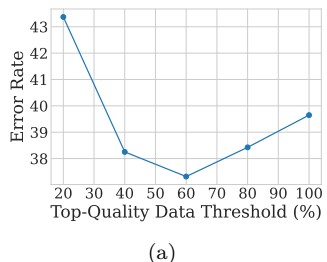

(a)

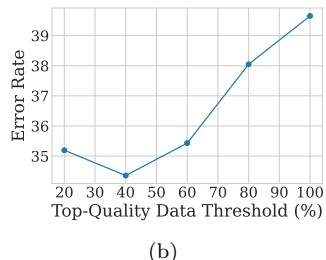

(b)

Figure 5: **Data Quality:** Zero-Shot Performances on ImageNet. Figure 5(a) shows results trained for **one epoch**. Figure 5(b) shows results trained for **the same number of sampled data**. We use ViT-B/32 as the vision encoder and ViT-B as the text encoder.

**Zero-Shot** Figure 8(a) showcases the zero-shot performances on ImageNet after training for one epoch using various sizes of training sets with different ViT models. The results indicate that ViT-L/16 does not consistently outperform other ViT models. In fact, when the number of samples is less than 100 million, ViT-L performs the worst among the ViT family. However, as the size of sampled data increases, larger vision encoders exhibit better performances. We conduct experiments on different datasets for an equal number of iterations, keeping the number of sampled data constant. Our results reveal that a larger dataset yields better performance, even when the number of iterations is held constant. Notably, we observe that augmenting the dataset to 400M does not yield any significant improvement in zero-shot performance due to our computation limits. Moreover, we discover that the performance gap between larger ViTs and their smaller counterparts widens as the dataset size increases.

Our experiments also investigate the relationship between zero-shot performance on ImageNet and other distributional datasets. We find that the accuracy curves for ImageNet-R (Hendrycks et al., 2020), ImageNet-Sketch (Wang et al., 2019), ImageNet-V2 (Recht et al., 2019), and ObjectNet (Barbu et al., 2019) are nearly linear as shown in Figure 21 in Appendix, indicating that the image classification ability of CLIP models improves consistently across these datasets. However, the accuracy of ImageNet-A (Hendrycks et al., 2019) does not exhibit a linear relationship with that of ImageNet (Deng et al., 2009) as shown in Figure 21(a) in Appendix. Specifically, when ImageNet accuracy is below 50%, the accuracy of ImageNet-A increases only slightly, while it grows more rapidly for higher ImageNet accuracies. We attribute this behavior to the presence of more challenging test examples in ImageNet-A, which is not present in the other four datasets that only contain variants of ImageNet images. We also find that for most of OOD datasets, when the standard accuracy is same, a larger ViT will have better robustness performances.

**Linear Probing** As shown in Figure 10, we present the linear-probing results and observe that for smaller datasets, the performance of ViT-L/16 is inferior to that of the other three models. However, for larger numbers of sampled data, the larger ViT models deliver significantly better results.

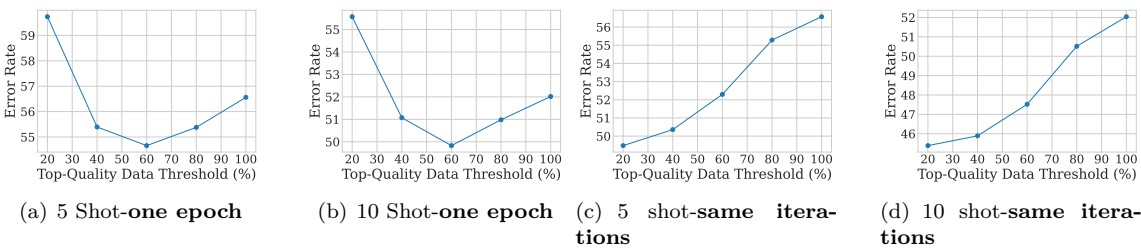

(a) 5 Shot-**one epoch**    (b) 10 Shot-**one epoch**    (c) 5 shot-**same iterations**    (d) 10 shot-**same iterations**

Figure 6: **Data Quality:** Few-Shot Performances on ImageNet. We train dataset for various quality thresholds for **one epoch** in (a) and (b). We train dataset for various quality thresholds for **the same number of sampled data** in (c) and (d).

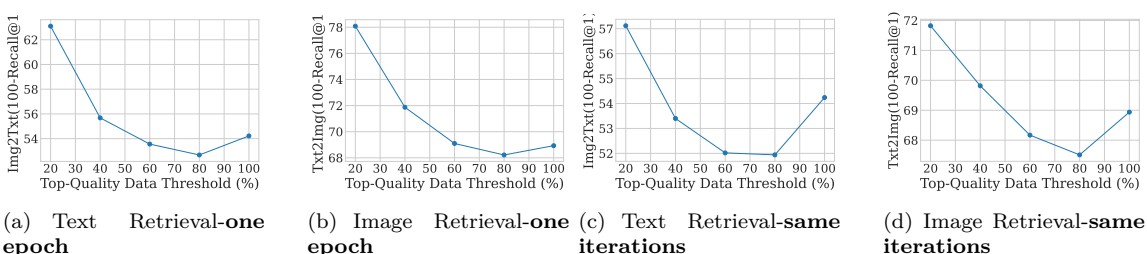

(a) Text Retrieval-**one epoch**    (b) Image Retrieval-**one epoch**    (c) Text Retrieval-**same iterations**    (d) Image Retrieval-**same iterations**

Figure 7: **Data Quality:** Retrieval Performances on MSCOCO. We train dataset for various quality thresholds for **one epoch** in (a) and (b). We train dataset for various quality thresholds for **the same number of sampled data** in (c) and (d)

Furthermore, in Figure 22 in Appendix, we compare ImageNet with other out-of-distribution (OOD) variants and find that the performance of vision encoders significantly declines for most OOD variants, except for ImageNet-V2.

Our results also show that for smaller ViTs, the effect of data diversity is smaller. For Ti/16, when the dataset size is larger than 25M, the increase of dataset size will not benefit the linear probing performance. However, for L/16, the increase of data diversity will benefit the linear probing performance until the training dataset size is larger than 400M. These findings demonstrate the importance of using large datasets for larger CLIP models.

**Retrieval** The results shown in Figure 11 demonstrate that ViT-L/16 exhibits poor performance for retrieval tasks when the number of sampled data points is less than 100M. However, with an increase in the number of sampled data points, larger ViT models demonstrate improved performance.

Furthermore, when the number of sampled data points is held constant, the trend in retrieval performances aligns with that of zero-shot performance. Moreover, as the dataset size increases, the performance of smaller ViT models plateaus earlier than that of larger models, suggesting that larger models derive greater benefits from additional data.

# 6 A Comparison of Network Architectures

Previous studies Zhai et al. (2021); Radford et al. (2021) have investigated different vision encoders for CLIP, including ResNet He et al. (2016), MLP-Mixer Tolstikhin et al. (2021), and ViT Dosovitskiy et al. (2020). However, architectures such as Swin-Transformer Liu et al. (2021)) and ConvNext Liu et al. (2022) have not been explored. Moreover, previous works have only compared network performance for only one training setting. In this section, we compare several CNN and vision transformer architectures with similar FLOPs, namely ViT-B/32, ResNet-50, ConvNext-T, Swin-T, and Mixer-B/32.

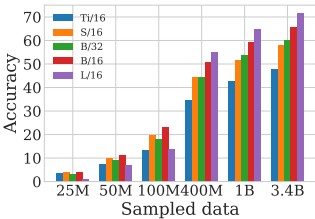
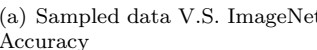

(a) Sampled data V.S. ImageNet Accuracy

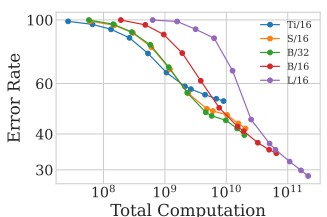

(b) Total Computation V.S. ImageNet Error Rate

Figure 8: **Various ViTs:** Zero-Shot performances with **various numbers of sample data**. We use various vision encoders and the same text encoder ViT-B. The total computation is computed by GFLOPs per sample times the number sampled data. The total computation is computed by GFLOPs per sample times the number sampled data.

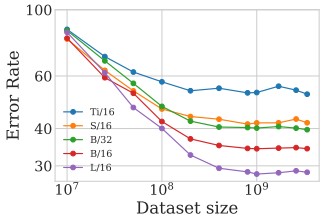

(a) Data size V.S. ImageNet Error Rate

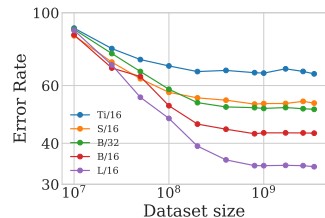

(b) Data size V.S. Average ImageNet Variants Error Rate

Figure 9: **Various ViTs:** Zero-Shot performances with **the same number of sampled data: 3.4B**. We use various vision encoders and the same text encoder ViT-B.

**Zero-Shot** We begin our investigation by examining the performance of different architectures for different numbers of sampled data. As illustrated in Figure 12(a), we observe that ResNet-50 outperforms ViT-B/32 when the number of sampled data is limited. However, as the number of samples increases, ViT-B/32 achieves better performance. One intuitive explanation is that CNNs possess a stronger inductive bias for images and can capture more local information even with a small number of samples. Moreover, we find that the performance of vision transformers variants does not show notable improvements when trained on a limited number of samples, and in most cases, Swin-T's performance is inferior to that of ViT-B/32. We plot the total computation cost versus error rate in Figure 12(b). When the total computation cost is low, ResNet-50 outperforms other architectures.

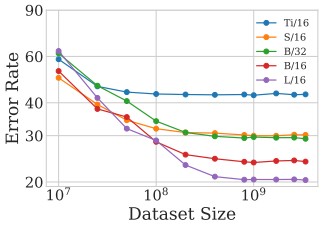

(a) Data size V.S. ImageNet

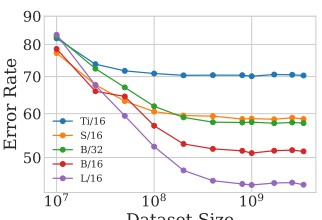

(b) Data size V.S. Average on OOD datasets

Figure 10: **Various ViTs:** Linear probing performances with various sizes of vision encoders. We train all subsets with **the same number of sampled data: 3.4B**.

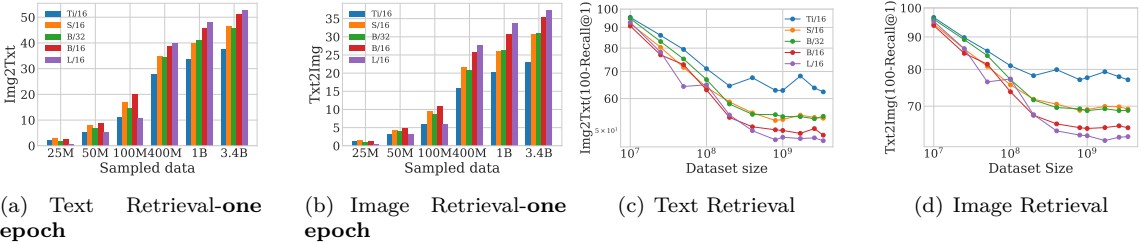



(a) Text Retrieval-**one epoch**
(b) Image Retrieval-**one epoch**
(c) Text Retrieval
(d) Image Retrieval

Figure 11: **Various ViTs:** Retrieval Performances on MSCOCO.

Next, we evaluate the effect of dataset size on the zero-shot performance of different architectures, while keeping the number of iterations constant. As demonstrated in Figure 13, ViT-B/32 consistently delivers the best performance. We observe that the impact of data diversity is similar across different networks, and increasing the dataset size beyond 400M does not offer any significant benefit to the zero-shot performance of any network. Therefore, the improvement in zero-shot performance due to data diversity is mainly related to model size rather than model architecture.

We also present the zero-shot performances of ImageNet and its variants in Figure 23. When the ImageNet accuracy is comparable, Swin-T and ViT-B/32 exhibit better performances on ImageNet variants, particularly on ImageNet-A. Our findings suggest that when the training dataset is sufficiently large, vision transformers provide a better choice as the vision encoder of CLIP.

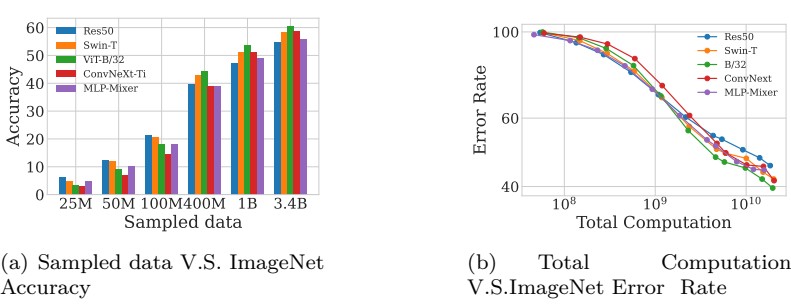

(a) Sampled data V.S. ImageNet Accuracy
(b) Total Computation V.S.ImageNet Error Rate

Figure 12: **Various architectures:** Zero-Shot performances with **various numbers of sampled data**. We train them for **one epoch**.

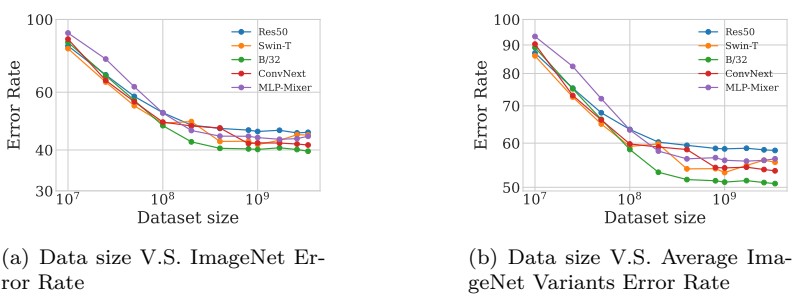

(a) Data size V.S. ImageNet Error Rate
(b) Data size V.S. Average ImageNet Variants Error Rate

Figure 13: **Various architectures:** Zero-Shot performances with **various dataset sizes** and **the same number of steps**.

**Linear Probing** To validate the performance of various encoders in CLIP models, we evaluate them using linear probing and present the results in Figure 14. It can be observed that when the number of sampled data is below 100 million, MLP-Mixer outperforms the other four architectures on both ImageNet and other OOD variants. However, as the number of sampled data increases, ViT-B/32 exhibits better results, particularly

on OOD variants. These results provide insights for selecting appropriate feature extractors for datasets with different sizes. Specifically, ResNet is a better choice when the number of sampled data is small, while ViT can achieve better robustness and standard accuracy when the number of sampled data is sufficiently large.

In Figure 24 in Appendix, we present the results of linear probing on ImageNet and various OOD datasets. Notably, we observed a significant drop in performance on ImageNet-A, -R, and ObjectNet after linear probing, whereas the performances on ImageNet-V2 and -Real improved. When the standard accuracy is close, we noticed that ViT and MLP-Mixer demonstrate better robustness. We hypothesize that this might be due to the relatively lower inductive bias of these two architectures, meaning they are less prone to overfitting to the specific dataset they were trained on. This can lead to better generalization and performance on out-of-distribution datasets.

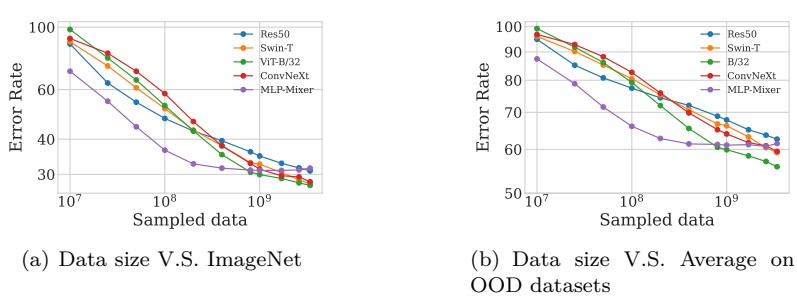

(a) Data size V.S. ImageNet

(b) Data size V.S. Average on OOD datasets

Figure 14: **Various architectures:** Linear Probing with **the same number of sampled data**.

**Retrieval** When considering different numbers of sampled data points, ResNet-50 demonstrates superior performance for smaller sample sizes, as depicted in Figures 15(a) and 15(b). However, as the number of sampled data points exceeds 400M, ViT-B/32 surpasses ResNet-50 in performance for both few-shot and retrieval tasks.

Moreover, when the number of sampled data points is consistent, Mixer-B/32 exhibits the poorest performance for retrieval tasks, as shown in Figures 15(c) and 15(d). These results indicate that ViT is the preferred choice for the vision encoder not only in zero-shot and linear probing tasks but also in few-shot and retrieval tasks.

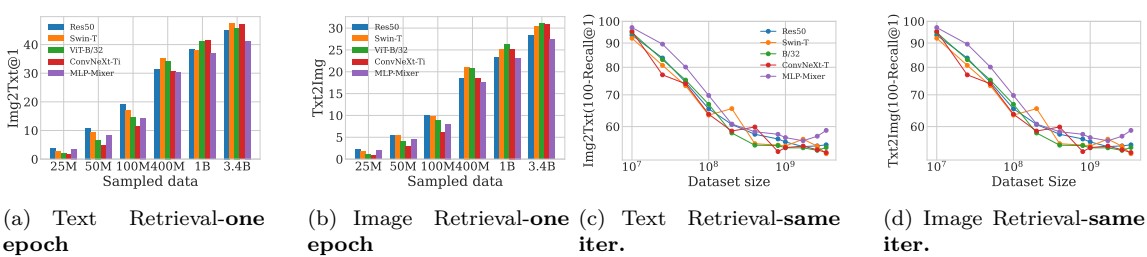

(a) Text Retrieval-**one epoch**

(b) Image Retrieval-**one epoch**

(c) Text Retrieval-**same iter.**

(d) Image Retrieval-**same iter.**

Figure 15: **Various Architectures:** Retrieval Performances on MSCOCO.

# 7 Training Strategies

In this section, we compare the training strategies of SLIP (Mu et al., 2021) and FLIP (Li et al., 2022) with the original CLIP. We use ViT-B/32 as the vision encoder and ViT-B as the text encoder for all three strategies. We follow the implementation of SLIP as described in Mu et al. (2021). For FLIP (Li et al., 2022), we apply a 50% patch masking and double the batch size. We also propose our own training strategy, *CLIP+Data Augmentation*.

**CLIP + Data Augmentation** While Mu et al. (2021) proposed enhancing the vision encoder of CLIP models through self-supervised learning, this comes at a significant computation cost compared to the original CLIP. To achieve similar enhancements to the vision encoder while saving on computation, we propose applying data augmentation to input images of CLIP. We validate the effectiveness of this approach on four subsets, training them for 30 epochs to ensure model convergence. We apply three data augmentation methods: crop&flip, RandAugment (Cubuk et al., 2019), and Stacked RandAugment (Khosla et al., 2020). The results, shown in Figure 1(b), demonstrate that all three methods improve performance compared to raw CLIP, highlighting the effectiveness of data augmentation in CLIP. Moreover, data augmentation on input images does not bring any extra computation cost compared to CLIP. Even when trained on a dataset of 25 million samples, the Stacked RA model can achieve comparable performance to that obtained with a dataset of 50 million samples.

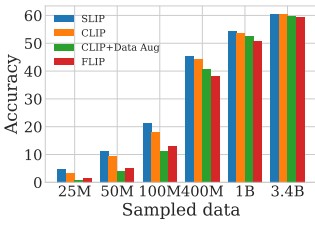

(a) Sampled data V.S. ImageNet Accuracy

(b) Total Computation V.S. ImageNet Error Rate

Figure 16: **Various Training Stategies:** Zero-Shot performances with **various numbers of sampled data**. We all subsets for **one epoch**. We use ViT-B/32 as the vision encoder and ViT-B as the text encoder for all strategies.

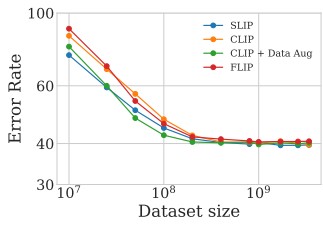

(a) Data size V.S. ImageNet

(b) Data size V.S. Average on ImageNet Variants

Figure 17: **Training Stategies:** Zero-Shot performances with various subsets while **the same number of sampled data**. We use ViT-B/32 as the vision encoder and ViT-B as the text encoder for all strategies.

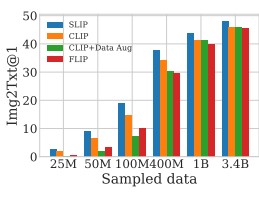

(a) Text Retrieval-**one epoch**

(b) Image Retrieval-**one epoch**

(c) Text Retrieval-**same iterations**

(d) Image Retrieval-**same iterations**

Figure 18: **Training Strategies:** Retrieval Performances on MSCOCO.

**Zero-Shot** Our experimental results on the ImageNet dataset, as depicted in Figure 16(a), demonstrate that SLIP outperforms the other two strategies when the number of training samples seen by models is less than one billion. This suggests that self-supervised learning for the vision encoder can be beneficial when the

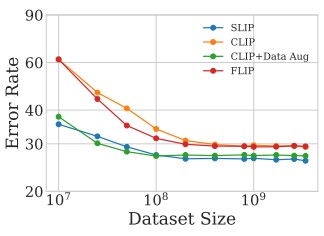

(a) Data size V.S. ImageNet

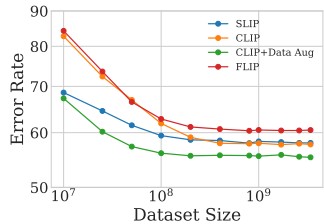

(b) Data size V.S. Average on OOD datasets

Figure 19: **Various Training Stategies:** Linear probing performances with various subsets while **the same number of sampled data**.

number of training samples is limited. However, as the number of training samples increases, both CLIP and FLIP achieve similar results and outperform SLIP. These results indicate that enhancing the vision encoders may not be necessary when the training dataset is large. Notably, when the model architecture is the same, SLIP is two times more computationally expensive than CLIP. As illustrated in Figure 16(b), when the total computation cost is the same, SLIP has the worst zero-shot performance on ImageNet. We also observed that data augmentation hurts the zero-shot performance when we only train these subsets for one epoch.

Moreover, we investigated how the four training strategies perform with various dataset sizes, with a fixed number of sampled data. As shown in Figure 17, SLIP outperforms CLIP and FLIP when the dataset size is small on ImageNet. However, as the dataset size increases, SLIP does not show better performance on ImageNet variants. On the other hand, CLIP + Data Aug performs better than the other three training methods, especially on ImageNet variants. Our results suggest that data augmentation on input images can bring better generalization performance with almost no extra cost.

In Figure 26 in Appendix, we present the results of our study on ImageNet compared to various OOD variants. SLIP also does not show better performance on these variants when the ImageNet accuracy is the same. However, CLIP + Data Aug brings better performance on ImageNet-Sketch and -R. In conclusion, we can say that CLIP + Data Aug can bring better zero-shot performance on both ImageNet and its variants when we train multiple epochs for the dataset.

**Linear Probing** We evaluated vision encoders trained with various strategies using linear probing and found that those trained with CLIP + Data Aug consistently outperform the other three strategies across all data points, particularly on OOD datasets as shown in Figure 19. The performances of ImageNet and OOD variants are compared in Figure 27 in Appendix, where CLIP and CLIP + Data Aug show better robustness than SLIP when ImageNet accuracy is the same. Considering the linear probing performances and computation cost, we believe that combining CLIP and data augmentation can result in a better feature extractor.

**Retrieval** The performance of SLIP, CLIP+Data Aug, CLIP, and FLIP is evaluated for few-shot and retrieval tasks on ImageNet with varying numbers of sampled data. Moreover, SLIP consistently outperforms the other methods on both image and text retrieval tasks as shown in Figures 18(a) and 18(b).

When the number of sampled data is fixed, the performance of each method for few-shot and retrieval tasks shows a different trend compared to the zero-shot performance. For retrieval tasks, SLIP consistently performs best across all dataset sizes as shown in Figures 18(c) and 18(d). These results suggest that SLIP is a better strategy for retrieval tasks, although it may not be the best one for classification tasks.

## 8 Conclusion

In this paper, we investigate the impact of data size, network architecture, and training strategies on the performance of CLIP. Our experiments demonstrate the importance of data quantity and data quality. We also show that data augmentation techniques can enhance the performance of CLIP without significantly

increasing the computational cost. Additionally, we explore the impact of different network architectures and training strategies on the performance of CLIP. Our results suggest that some architectures and training strategies perform better than others at different computational budgets, highlighting the importance of careful selection.

## 9    Acknowledgements

We acknowledge Emirhan Kurtulus for helpful discussions. We also acknowledge Douglas Eck, Andrew Pierson, and the rest of the Google DeepMind team for their support.

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
