# A Appendix

## A.1 Data

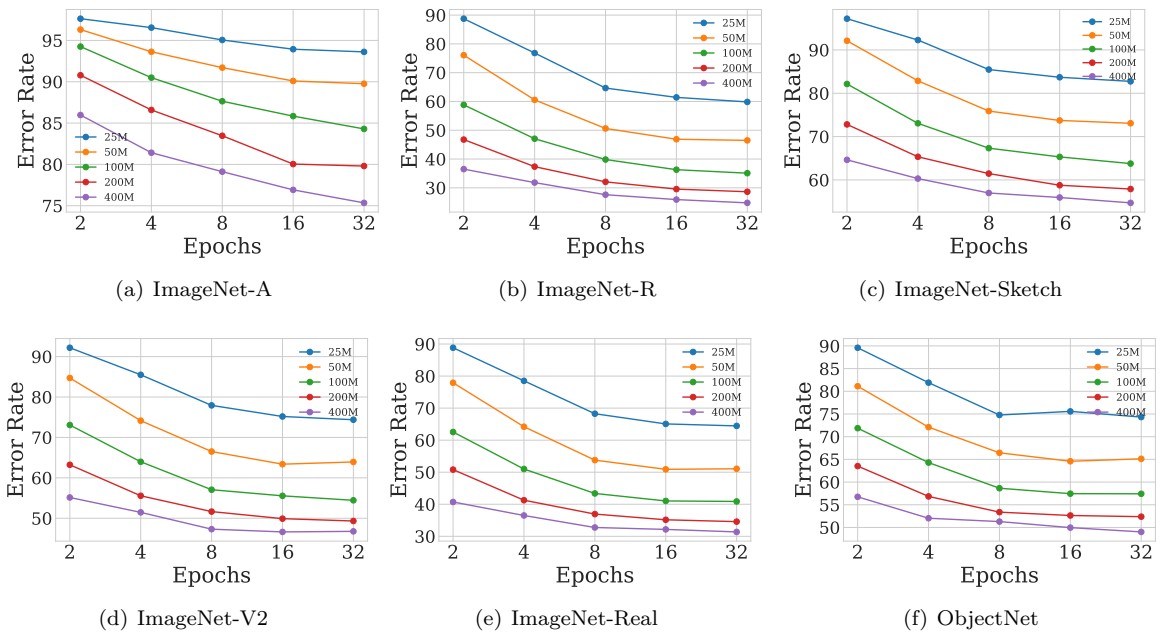

(a) ImageNet-A     (b) ImageNet-R     (c) ImageNet-Sketch

(d) ImageNet-V2     (e) ImageNet-Real     (f) ObjectNet

Figure 20: **Data Quantity:** Zero-Shot performances on ImageNet V.S. ImageNet variants.

## A.2 Variants of Vision Transformers

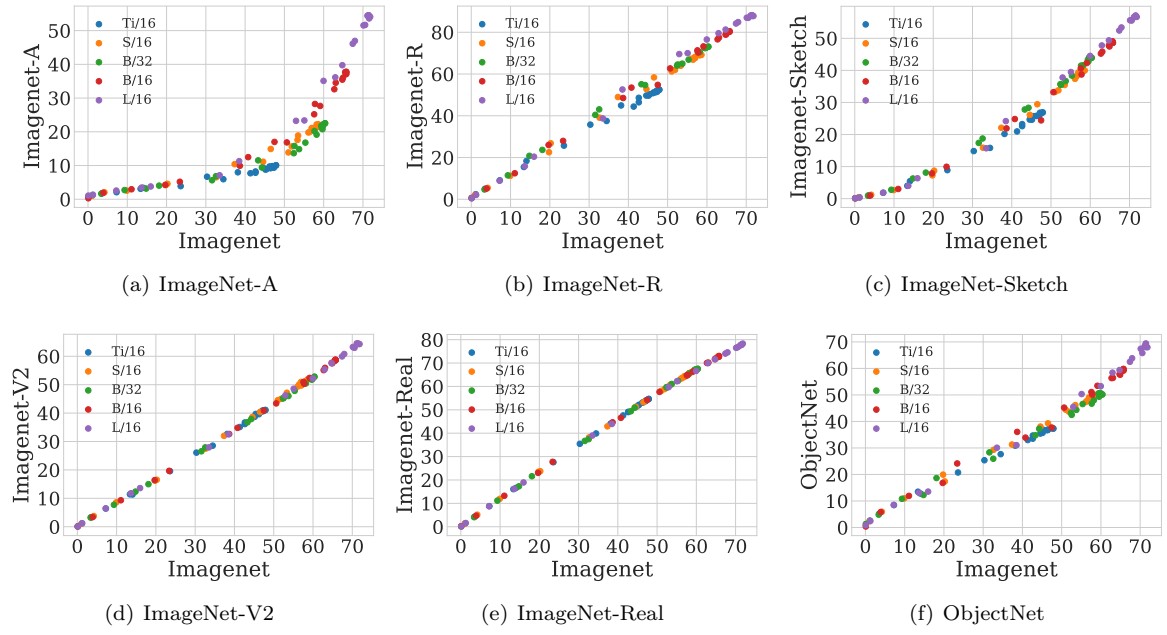

(a) ImageNet-A     (b) ImageNet-R     (c) ImageNet-Sketch

(d) ImageNet-V2     (e) ImageNet-Real     (f) ObjectNet

Figure 21: **Various ViTs:** Zero-Shot performances on ImageNet V.S. ImageNet variants.

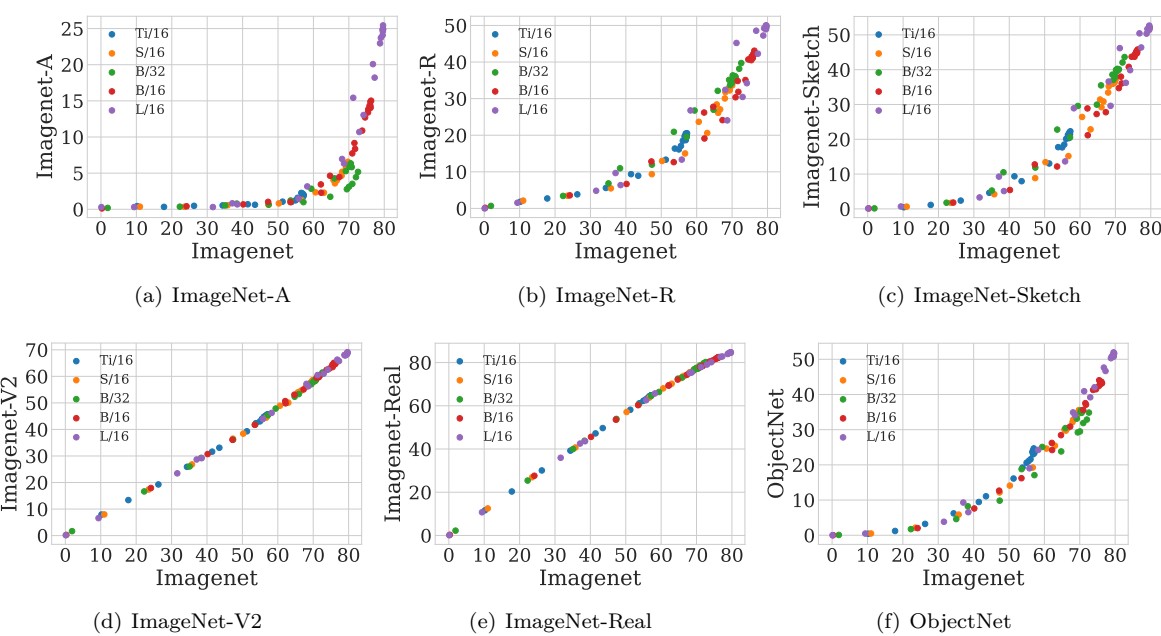

Figure 22: **Various ViTs:** Fine-Tuning performances on ImageNet V.S. OOD datasets

## A.3    Various Network Architectures

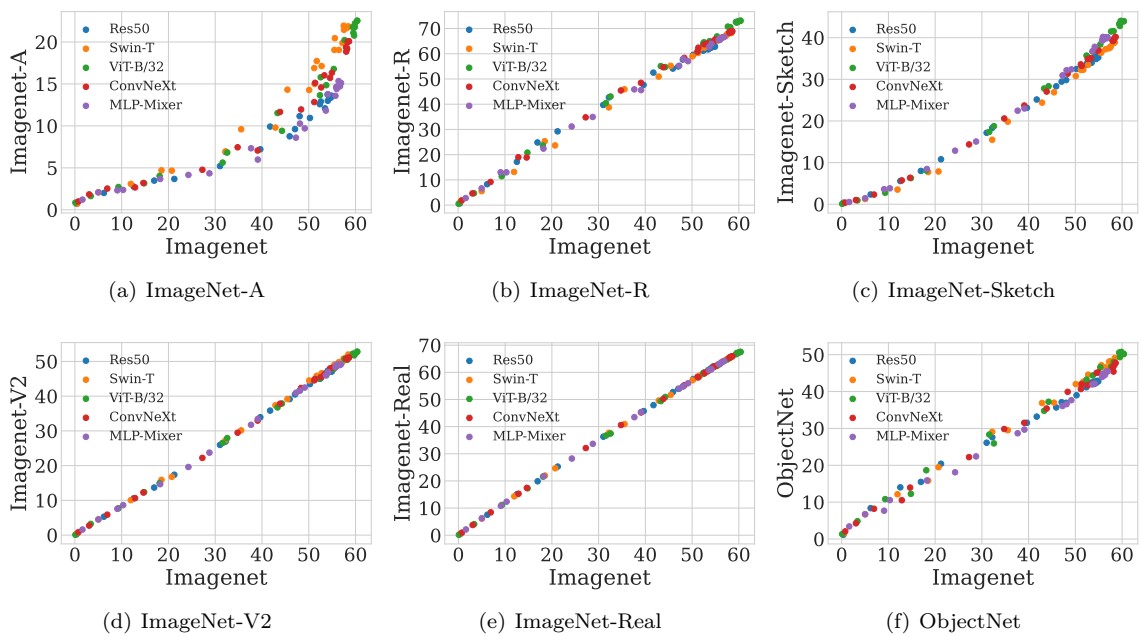

Figure 23: **Various architectures:** Zero-Shot performances on ImageNet V.S. ImageNet Variants

## A.4    Training Strategies

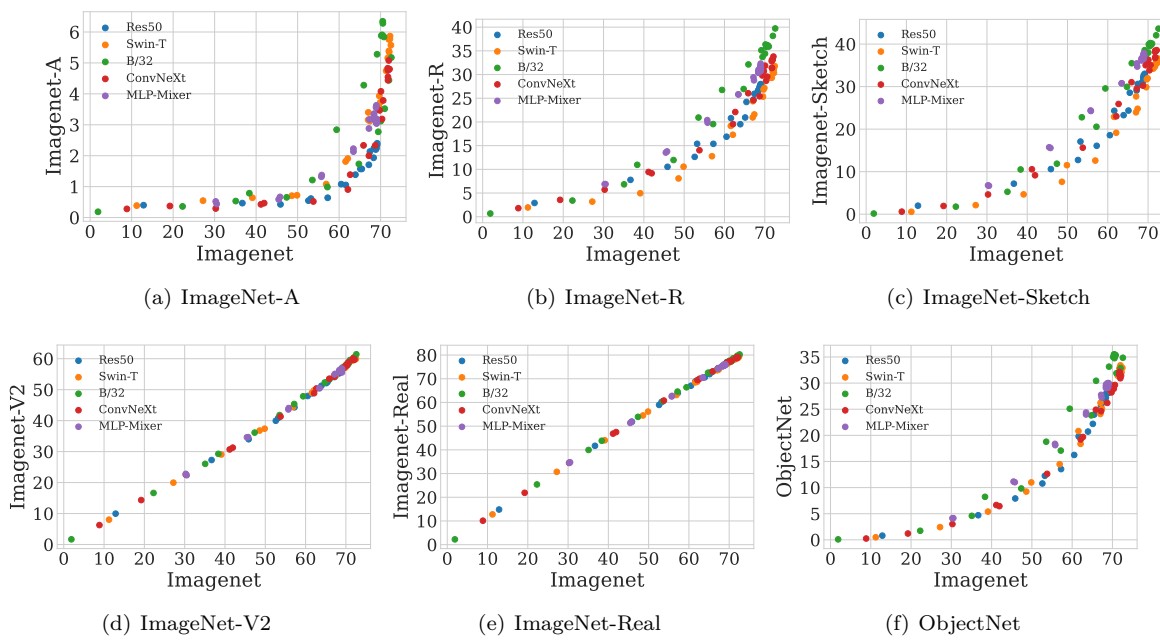

(a) ImageNet-A     (b) ImageNet-R     (c) ImageNet-Sketch

(d) ImageNet-V2     (e) ImageNet-Real     (f) ObjectNet

Figure 24: **Various architectures:** Linear probing performances on ImageNet V.S. zero-shot performances on OOD datasets

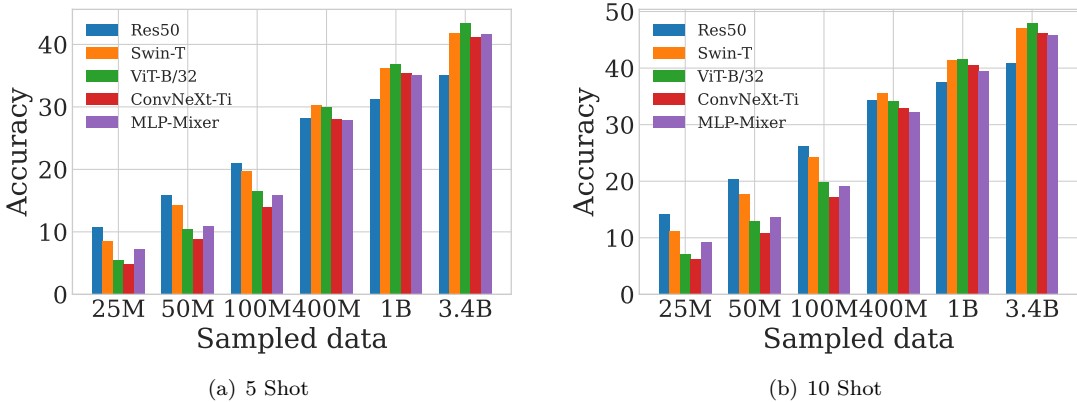

(a) 5 Shot        (b) 10 Shot

Figure 25: **Various Architectures:** Few-Shot Performances on ImageNet with **various numbers of sampled data**.

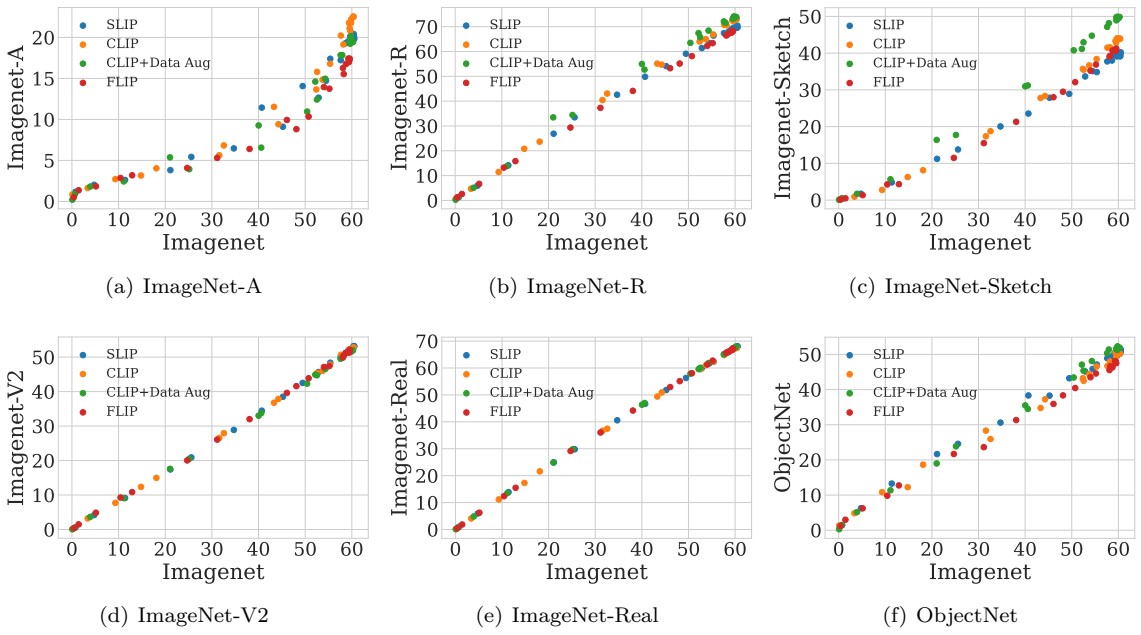

Figure 26: **Various Training Stategies:** Zero-Shot performances on ImageNet V.S. ImageNet Variants.

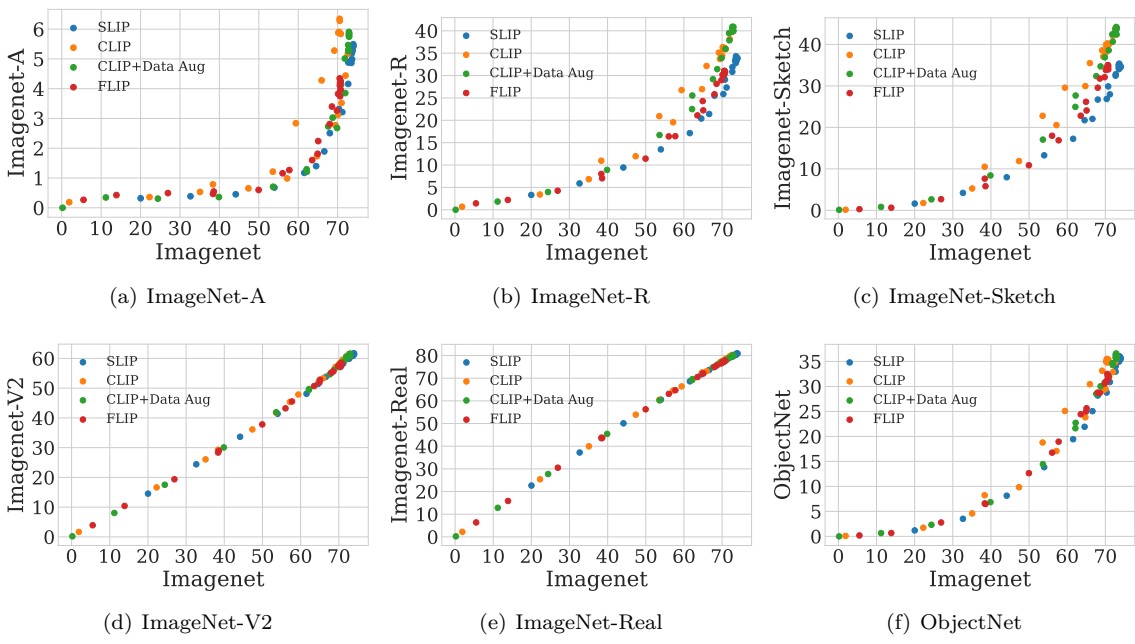

Figure 27: **Various Training Stategies:** Linear probing performances on ImageNet V.S. OOD datasets