# OpenReview forum: "Scaling (Down) CLIP: A Comprehensive Analysis of Data,Architecture, and Training Strategies"
_TMLR — Accepted by TMLR_

### Review · Reviewer_MqEg · 2023-11-27

**Summary Of Contributions:**

This work provides an in-depth exploration of CLIP (and variants, such as FLIP and SLIP) in a variety of settings, varying the model architecture, training iterations, and number of training samples filtered according to an image-caption similarity threshold. This work also proposes adding image augmentations in the CLIP pretraining pipeline, namely a stacked RandomAugment strategy, and sees performance improvements in certain settings. This work primarily focuses on gathering insights in settings with small training data, small models, and few training iterations.

**Audience:**

Yes

**Claims And Evidence:**

Yes

**Requested Changes:**

*Suggestion 1:*
Figure 2a: increasing training epochs for smaller datasets does not significantly decrease performance, but increasing training epochs for larger datasets is important. However, I actually observe the opposite trend in this figure. Relative performance drop on the 400M datasets is about 9% whereas relative drop on the 25M datasets is about 21%. What I do see is that their performance plateaus from 16 to 32 epochs, but this is not the case for the 40M dataset. I would suggest finding a better way to illustrate/visualize this result.

*Suggestion 2:*
Text at the bottom of Page 4 stops suddenly and does not continue into page 5. Similarly, on Page 5, the text in the first paragraph stops suddenly, and the conclusions from Figure 5 (b) are never discussed. Please correct these issues.

*Suggestion 3:*
Figure 6 (c)/(d) regarding data quality is particularly interesting, however many details are absent from this experiment, such as the number of iterations which the models are trained for. I would expect this specific hyper-parameters have a significant impact on the observations. For instance, with much longer training schedules would all methods perform similarly? Are these observations mostly concerning short training schedules, in which case, the observations are that high quality data yield faster convergence, but not better performance? Or do the models trained on smaller filtered data end up performing better with longer training as well?

*Suggestion 3:*
Observation that larger datasets yield better performance, even when the number of iterations is held fixed is interesting; which is mentioned at the start of page 7. But which experiments support this result, as such a conclusion is not evident from Figure 8, which trains for one epoch on each dataset size, implying that larger datasets training for more iterations. I believe Figure 9 a page down may be demonstrate this result, but I would suggest clarifying this in the text.

*Suggestion 4:*
It is claimed that ConvNeXt has not been explored, but I would like to point out other the authors that the ConvNeXt has been thoroughly explored in OpenCLIP. Can you please clarify where that exploration falls short?

*Suggestion 5:*
Confused by Figure 16 (a) which seems to contradict figure 1b and the claim that CILP+data aug outperforms CLIP. In figure 16 (a) vanilla CLIP outperforms CLIP + DataAug for all randomly sampled data scales. I would also expect the ability to get improvements in performance to depend heavily on the data; as aggressive image transformations can result in high misalignment between the caption and the image. This also contradicts the observations in Figure 17. Can you please clarify these points, and provide more insight into this relationship?

**Strengths And Weaknesses:**

**Strengths**
* Besides some issues where text appears to be missing/incomplete, this work is relatively clear and provides some intriguing insights into the behaviour of CLIP, and the impact of a wide variety of design choices in certain resource capped settings.
* This work will be of interest to a wide audience given the broad applicability and use of CLIP features in the literature.
* This work complements existing scaling results on CLIP by focusing on settings where the number of training iterations, model sizes, data size, etc. is small. It is akin to an inverse scaling ablation of CLIP.

**Weaknesses**

I provide detailed suggestions/comments on the manuscript in the other sections below, but the major weakness in my opinion is the lack of a coherent story or messaging to the work. While this work covers a broad range of ablations, it does not dive deeper into any of them. Rather, it feels as if there are many interesting one-off experiments that have been stitched together into a technical report, and noticeably lacking is an in-depth study of any one of these ablated factors.

For instance, Figure 6 (c)/(d) regarding data quality is particularly interesting, but why does similarity thresholding on the image-caption pairs lead to better 5-shot or 10-shot IN1k performance? Are there other ways of measuring data quality? What training schedule is used in Figure 6 (c)/(d) and how would the findings change with different schedules (longer vs shorter)?

Another example would be diving deeper into the model architecture experiments. For instance, being able to provide measurable experiments that explain why a RN50 would outperform a ViT-B/32 with limited data and training iterations would be helpful beyond simply recording this observation.

In short, it would be more helpful to dive deeper into any of these experimental settings and formulate and test hypotheses about the expected behaviour of models in such settings, rather than simply recording results in various settings. In my opinion, a depth rather than breadth approach would be particularly helpful for improving the impact of this work.

---

> ### Author Response · Authors · 2024-01-30
> **Re: Review of Paper1702 by Reviewer MqEg**
>
> Q: The lack of a coherent story or messaging to the work.
>
> A: The core objective of our work is to systematically explore factors influencing CLIP training and performance, aiming to derive optimal training strategies under various scenarios, such as small data size and constrained computation costs. By investigating data size, model size, and training strategies, closely tied to computation cost, we strive to offer valuable insights to the community. We appreciate the recognition of the interesting results presented in this work. To provide a more cohesive narrative, we intend to delve deeper into these aspects in subsequent works, ensuring a more comprehensive and focused storyline.
>
>
>
> Q: Why does a smaller thresholding on the image-caption pairs lead to better 5-shot or 10-shot IN1k performance? (Figure 6(c) (d))
>
> A: In Figure 6, the x-axis represents the Top-Quality Data Threshold, where a smaller number signifies a higher-quality dataset. The correlation score between image-text pairs measures dataset quality, with a lower score indicating a closer relationship between the image and text. Therefore, a smaller threshold leads to better 5-shot or 10-shot IN1k performance as it corresponds to a higher-quality dataset, enhancing the correlation between image and text pairs.
>
> Q: why a RN50 would outperform a ViT-B/32 with limited data and training iterations
>
> A: The inductive bias of CNNs, characterized by locality and translation equivariance, implies that CNNs possess a substantial amount of prior knowledge. This results in the need for less data to effectively train a model.
>
> Q: Clarification for Figure 2(a)
>
> A: In Figure 2(a), the y-axis denotes the error rate, with a lower error rate indicating better performance. The rapid decrease in error rate for the 25M data size training, as we increase the number of training epochs, is attributed to the initially high error rate. Conversely, as the dataset size grows, the error rate of lower training epochs decreases. Therefore, the error rate of the 400M dataset does not drop as fast as the 25M dataset when increasing the number of training epochs.
>
>
> Q: Details about data quality
>
> A: We appreciate the insightful question. To evaluate the effect of data quality, we employed two training schedules. The first involves training datasets of various qualities for a single epoch, with the number of iterations for the top 20% dataset being 20% of the total iterations. The second schedule trains datasets with varying qualities for an equal number of iterations, as illustrated in Figures 5(a), 6(a)(b), 6(c)(d). Our findings highlight that, even with fewer training iterations, better-quality datasets outperform poorer-quality counterparts, underscoring the impact of data quality on CLIP performance. Additionally, when the number of training iterations is consistent, superior data quality contributes to improved zero-shot and few-shot performances, affirming the importance of balancing data quality and diversity.
>
>
>
> Q: Large dataset yields better performance when the number of iterations is the same
>
> A: Thank you for bringing up this point. Figure 9(a)(b) demonstrates this observation. By selecting datasets of varying sizes and training them for an equal number of iterations, we consistently observe a decrease in error rate as the dataset size increases. This supports the notion that larger datasets lead to better performance under equivalent training conditions.
>
>
>
> Q: Exploration about ConvNext
>
> A: We appreciate the inquiry. At the time of writing our paper, ConvNext had not been investigated. While we acknowledge OpenCLIP's exploration of ConvNext, our work provides additional details, including performances of ConvNext with various dataset sizes and a thorough comparison with other network architectures. This enhances the understanding of ConvNext's capabilities in diverse scenarios.
>
> Q: CLIP + data aug & CLIP
>
> A: Thank you for the insightful question. In Figure 16, the superior zero-shot performances of CLIP compared to CLIP + data aug can be attributed to our training setup, where all datasets are trained for only one epoch. In such a scenario, data augmentation does not provide substantial benefits. However, in Figure 16, where the number of iterations is fixed for all datasets, and smaller datasets receive more training epochs, CLIP + data aug outperforms CLIP significantly. This underscores the effectiveness of data augmentation, especially when applied over extended training periods.

---

### Review · Reviewer_qMQC · 2023-12-25

**Summary Of Contributions:**

This paper examines various training configurations for CLIP. Specifically, the paper reveals that 1) small high-quality data is better than large low-quality data, 2) small ViTs perform better than large ones with smaller datasets, 3) CNNs are better than ViTs in small-scale regimes, 4) SLIP and data augmentation improve CLIP but lead to slower convergence.

**Audience:**

Yes

**Claims And Evidence:**

Yes

**Requested Changes:**

- Emphasize the findings specific to vision-language.
- Address the editorial comments.

**Strengths And Weaknesses:**

## Strength

- Extensive analysis of various training configurations for CLIP.
- Enormous computational resources are utilized for this training.


## Weakness

- Additional relevant works
    - This paper reminds me of the analysis papers like [1,2]. It would be great to discuss their findings and clarify the differences from this work. For example, [1] studies ViT in the vision-only setup, while this paper focuses on vision-language training. If different trends emerge between vision-only and vision-language setups, it would be worth highlighting them.
    - A minor point, but there are more works on scaling down CLIP [3].
    - Another minor point, but the effect of data quality is studied in recent data curation works [4,5].

[1] Steiner et al. "How to Train Your ViT? Data, Augmentation, and Regularization in Vision Transformers." TMLR 2022.\
[2] Hendricks et al. "Decoupling the Role of Data, Attention, and Losses in Multimodal Transformers." TACL 2021.\
[3] Li et al. "An Inverse Scaling Law for CLIP Training." NeurIPS 2023.\
[4] Xu et al. "CiT: Curation in Training for Effective Vision-Language Data." arXiv 2023.\
[5] Xu et al. "Demystifying CLIP Data." arXiv 2023.

- Observations are not surprising
    - It is intuitive that smaller ViTs and CNNs are better for smaller datasets. Also, SLIP and CLIP + data augmentation would improve CLIP but can lead to slower convergence. Nevertheless, I think the analyses are worth reporting, especially the comparisons with CLIP, SLIP, and FLIP. It would be more interesting to see other training strategies like CoCa and BLIP though.
    - As mentioned above, if the paper highlights the findings that are specific to vision-language, it would be more exciting.

- Considerations on data augmentation for CLIP
    - As far as I know, a tricky aspect of applying data augmentation to CLIP is that the augmentation may disrupt the alignment of images and text. For example, RandomCrop may remove some objects from the caption, which still remains in the caption. Can the authors comment on this?

- Editorial comments
    - Check the "data quality" part in Section 4. It seems that the paragraphs after "%" are commented out. Also, please maintain consistent usage of "top 20" vs. "Top 40".
    - Some \citep references are not spaced from the previous words. Check pages 5 and 6, for example.
    - Figures are too small. Maybe the authors can consider increasing the figure size and placing less important ones in the Appendix.
    - TMLR can merge the Appendix into a single PDF, which would enhance readability.

---

> ### Author Response · Authors · 2024-01-30
> **Re: Review of Paper1702 by Reviewer qMQC**
>
> Q: Different trends between vision-only and vision-language setups.
>
> A: While we acknowledge shared sentiments with [1] on the positive impact of large models, it's essential to highlight the divergences in our respective approaches. Notably, [1] delved into the effects of specific data augmentation and regularization methods, whereas our focus primarily centers on the investigation of a singular data augmentation method—randaug. We deliberately exclude the exploration of regularization methods in our study, thereby emphasizing a distinct avenue of research.
>
>
>
> Q: More works on scaling down CLIP.
>
> A: Our study uniquely establishes a computation threshold for scaling down CLIP, specifically setting the total trained samples to be less than 3.4 billion. In this small regime, we study the  scaling-down law across various model sizes and architectures, further contributing novel insights to the field.
>
> Q: Observations are not surprising. Comparisons with BLIP etc.
>
> A: While previous works have indeed compared CNN and ViT for CLIP, our contribution lies in being the first to conduct such comparisons under diverse scenarios. Our experimentation spans small datasets with large training iterations to large dataset sizes with small training iterations, yielding  practical guidance for network architecture selection during CLIP training. BLIP and CoCa are outside the scope of current work.
>
> Q: Editorial comments.
>
> A: We sincerely appreciate the thoughtful editorial comments and assure the incorporation of the suggested modifications in the revised version of our paper.
>
>
>
> [1] Steiner, Andreas et al. “How to train your ViT? Data, Augmentation, and Regularization in Vision Transformers.” Trans. Mach. Learn. Res. 2022 (2021): n. Pag.

---

### Review · Reviewer_tNM9 · 2024-01-06

**Summary Of Contributions:**

The paper extensively explores CLIP training along three dimensions: training data, model architecture and training strategy, and provide several findings especially useful for training CLIP models with a limited computation budget. The most notable findings from experiments on zero-shot transfer, linear probing, few-shot classification and image-text retrieval are as follows:

- Dataset quality is quite important: training on a smaller dataset of high quality outperforms training on a larger dataset of lower quality.
- We should increase the dataset size for fully benefitting from increasing the ViT model size.
- CNN models perform better with a smaller dataset due to its high inductive bias while ViTs perform better with a larger dataset.
- While SLIP performs better when training on a smaller number of samples compared to FLIP and CLIP (+ augmentation), CLIP with strong data augmentation performs best when training on a larger dataset, except in the task of retrieval: SLIP performs best across all dataset sizes on image-text retrieval.

These findings will provide guidance on how to train CLIP models.

**Audience:**

Yes

**Broader Impact Concerns:**

There are no specific ethical issues regarding the paper, but please add the broader impact of the work.

**Claims And Evidence:**

Yes

**Requested Changes:**

Please modify the draft to reflect what I mentioned above. Also, it would be very good to compare different architectures, training strategies and model sizes when training on datasets of different distribution: e.g., sampling the same number of data from WebLI and LAION-5B [1].

Minor request: the paper addresses training CLIP models on a wide range of data scales, even beyond the original dataset scale (400M). Thus, I recommend changing the title of the paper: it is too narrow compared to the broader scope the paper actually addresses.

[1] Schuhmann et al. LAION-5B: An open large-scale dataset for training next generation image-text models. In NeurIPS Datasets and Benchmarks Track, 2022.

**Strengths And Weaknesses:**

Strengths
- The findings from the paper provides a comprehensive guideline on how to train CLIP models considering a broad range of factors affecting model performance.

The following can be both the advantage and disadvantage of this work:
- Even though the authors set the maximum number of samples, conducting all the experiments reported in the paper requires a huge amount of training costs that only a very few labs can replicate them. Unless the authors release all training checkpoints, it is very hard to validate the experimental results.

Weaknesses: overall, writing should be improved. There are incorrect or missing details mentioned in the paper.
- It seems that the paragraphs after "containing the top 20" and "the Top40" are commented out in Section 4.
- The original CLIP uses a relatively small text encoder while this work uses the same size as the vision encoder with very shot token length (16). Is there any reason for this?
- Please add the details of each of six ImageNet variants.
- In Section 4, the authors said that longer training epochs led to enhanced performance on ImageNet only with large datasets. On the contrary, however, the performance gain on longer training decreases as the dataset size increases according to Figure 2.
- In Section 5, the authors said that they sampled ten subsets of varying sizes from WebLI, and all the subsets had the same data distribution and quality. Please add the details about how to ensure the same data distribution and quality.
- The authors used the term "diversity" in Section 5 and Section 6: they said the diversity increased when the number of sampled data points increased. However, this is not true because all the subsets have the same data distribution.
- In Section 6, the authors said that ResNet-50 outperformed the other architectures on linear probing when the number of sampled data is below 100 million, but this is not true: Mixer-B/32 is the best according to Figure 14.
- The authors mentioned the few-shot classification task when comparing different model architectures and training strategies, but the experimental results on the task are missing.

---

> ### Author Response · Authors · 2024-01-30
> **Re: Review of Paper1702 by Reviewer tNM9**
>
> Q: Token length
>
> A: We appreciate the inquiry. The choice of a token length of 16 aligns with the setting established by [1]. Through offline experiments, we validated that varying token lengths do not significantly impact the final results. Thus, maintaining a token length of 16 enhances computational efficiency while ensuring consistency with the benchmark setting.
>
> Q: Confusion for Figure 2 and Section 4: Relationships between the number of epochs and datasize
>
> A: Thank you for bringing this to our attention. We recognize the need for clarification in our paper. To address the concern, we want to emphasize that, beyond a certain point, increasing the number of epochs does not notably enhance the performance of smaller datasets. Conversely, the performance of the 400M dataset continues to improve even as the number of epochs extends from 16 to 32. This insight underscores the nuanced relationship between epoch count and dataset size, and we will revise the corresponding sections for improved clarity.
>
> Q: Data distribution and quality of 10 subsets from WebLI
>
> A: We appreciate the question. The subsets from WebLI were randomly sampled to ensure comparable distribution and quality across all selected subsets. This random sampling strategy was implemented to mitigate biases and ensure the robustness of our findings.
>
> Q: Diversity in Sections 5 and 6
>
> A: Thank you for seeking clarification. In the context of our paper, diversity refers to the model's ability to process a broader range of data, with larger datasets containing more varied image-text pairs. We acknowledge the need for clearer articulation on this point and will enhance the explanation in the revised version to eliminate any ambiguity.
>
> Q: Comparison between ResNet-50 and MLP-Mixer on Linear Probing
>
> A: We appreciate your keen observation. The error in the comparison between ResNet-50 and MLP-Mixer on Linear Probing will be rectified in the next version of our paper. Thank you for bringing this to our attention.
>
> Q: Few-shot experiment results for different model architectures and training strategies
>
> A: Thank you for highlighting this omission. We acknowledge the importance of including few-shot experiment results for various model architectures and training strategies. In the next version, we will address this by incorporating the relevant data and analysis into the complementary material for a more comprehensive presentation of our findings.
>
>
> [1] Zhai, Xiaohua et al. “LiT: Zero-Shot Transfer with Locked-image text Tuning.” 2022 IEEE/CVF Conference on Computer Vision and Pattern Recognition (CVPR) (2021): 18102-18112.

---

### Decision · Action_Editor_dhn6 · 2024-03-02

**Recommendation:** Accept as is

**Comment:**

As the reviewers mentioned, the experiments are extensive and provides useful insights to understanding the behavior of CLIP. The empirical findings reported in this paper can be useful for future efforts on training CLIP-like models.

**Audience:**

This article is likely to be of interest to a broad audience within the TMLR community.

**Claims And Evidence:**

This paper offers valuable insights into the behavior of CLIP across various factors, including data, architecture, and training strategies, in resource-constrained settings. The reviewers agreed that the empirical results were presented in a clear and compelling manner. The authors have effectively addressed most of the initial concerns raised by the reviewers. Consequently, all three reviewers recommend accepting this article for publication.